# Angiopoietin-2 Inhibition of Thrombomodulin-Mediated Anticoagulation—A Novel Mechanism That May Contribute to Hypercoagulation in Critically Ill COVID-19 Patients

**DOI:** 10.3390/biomedicines10061333

**Published:** 2022-06-06

**Authors:** Michael Hultström, Karin Fromell, Anders Larsson, Barbro Persson, Bo Nilsson, Susan E. Quaggin, Christer Betsholtz, Robert Frithiof, Miklos Lipcsey, Marie Jeansson

**Affiliations:** 1Anaesthesiology and Intensive Care Medicine, Department of Surgical Sciences, Uppsala University, 751 85 Uppsala, Sweden; michael.hultstrom@mcb.uu.se (M.H.); robert.frithiof@surgsci.uu.se (R.F.); miklos.lipcsey@surgsci.uu.se (M.L.); 2Integrative Physiology, Department of Medical Cell Biology, Uppsala University, 751 23 Uppsala, Sweden; 3Department of Immunology, Genetics and Pathology, Uppsala University, 751 85 Uppsala, Sweden; karin.fromell@igp.uu.se (K.F.); barbro.persson@igp.uu.se (B.P.); bo.nilsson@igp.uu.se (B.N.); christer.betsholtz@igp.uu.se (C.B.); 4Department of Medical Sciences, Clinical Chemistry, Uppsala University, 751 85 Uppsala, Sweden; anders.larsson@akademiska.se; 5Cardiovascular Research Institute, Feinberg School of Medicine, Northwestern University, Chicago, IL 60611, USA; quaggin@northwestern.edu; 6Division of Nephrology and Hypertension, Northwestern University, Chicago, IL 60611, USA; 7Department of Medicine Huddinge, Karolinska Institutet, 141 52 Huddinge, Sweden; 8Hedenstierna Laboratory, CIRRUS, Anaesthesiology and Intensive Care Medicine, Department of Surgical Sciences, Uppsala University, 751 23 Uppsala, Sweden

**Keywords:** COVID-19, hypercoagulation, thrombomodulin, Angiopoietin-2, endothelial dysfunction

## Abstract

Hypercoagulation and endothelial dysfunction play central roles in severe forms of COVID-19 infections, but the molecular mechanisms involved are unclear. Increased plasma levels of the inflammatory cytokine and TIE2 receptor antagonist Angiopoietin-2 were reported in severely ill COVID-19 patients. In vitro experiments suggest that Angiopoietin-2 bind and inhibits thrombomodulin. Thrombomodulin is expressed on the luminal surface of endothelial cells where it is an important member of the intrinsic anticoagulant pathway through activation of protein C. Using clinical data, mouse models, and in vitro assays, we tested if Angiopoietin-2 plays a causal role in COVID-19-associated hypercoagulation through direct inhibition of thrombin/thrombomodulin-mediated physiological anticoagulation. Angiopoietin-2 was measured in 61 patients at admission, and after 10 days in the 40 patients remaining in the ICU. We found that Angiopoietin-2 levels were increased in COVID-19 patients in correlation with disease severity, hypercoagulation, and mortality. In support of a direct effect of Angiopoietin-2 on coagulation, we found that injected Angiopoietin-2 in mice associated to thrombomodulin and resulted in a shortened tail bleeding time, decreased circulating levels of activated protein C, and increased plasma thrombin/antithrombin complexes. Conversely, bleeding time was increased in endothelial-specific Angiopoietin-2 knockout mice, while knockout of Tie2 had no effect on tail bleeding. Using in vitro assays, we found that Angiopoietin-2 inhibited thrombomodulin-mediated anticoagulation and protein C activation in human donor plasma. Our data suggest a novel in vivo mechanism for Angiopoietin-2 in COVID-19-associated hypercoagulation, implicating that Angiopoietin-2 inhibitors may be effective in the treatment of hypercoagulation in severe COVID-19 infection.

## 1. Introduction

SARS-CoV-2 infection may be paucisymptomatic or lead to coronavirus disease-2019 (COVID-19), which has a wide range of symptoms and may cause severe illness, in particular in individuals with other cardiovascular risk factors [1,2]. Thrombotic and thromboembolic disease have emerged as major COVID-19 complications, despite routine thrombosis prophylaxis being the standard of care [3,4,5]. Micro-thrombosis was suggested as contributing to both respiratory failure and neurological complications [6,7], and the activation of the coagulation system indicates a poor prognosis among COVID-19 patients in intensive care [2,6,8,9].

Reports suggests that critically ill COVID-19 patients have severe vascular inflammation and endothelial injury, resulting in an imbalance between anticoagulation and thrombo-inflammatory mediators [10]. The switch to a procoagulant state of endothelial cells appears to be an important disease mechanism. Severe COVID-19 is associated with increased levels of procoagulant factors, including Von Willebrand factor (VWF) [11,12], vascular cell adhesion markers [12,13,14,15], increased thromboelastography (TEG) parameters [4,16,17], and increased circulating soluble thrombomodulin (sTM) [12,18,19], which is a cleaved product of the full length thrombomodulin (TM) from the endothelial surface during inflammation [20]. 

Angiopoietin-2 (ANGPT2) is an inflammatory cytokine, and circulating levels correlate with adverse outcomes in several critical care syndromes, including acute respiratory disease syndrome (ARDS) and sepsis, reviewed in [21]. Elevated plasma ANGPT2 is a strong predictor of death in infection-mediated ARDS, independent of the infectious agent, and elevated plasma ANGPT2 is associated with disseminated intravascular coagulation in conjunction with sepsis [22]. In COVID-19, recent data show that ANGPT2 levels correlate with the severity of disease [12,15,19,23,24]. 

ANGPT2 exerts its effects through several molecular mechanisms, the most well-studied being inhibition of TIE2 receptor signaling, that normally remains activated by ANGPT1 secreted from perivascular cells. ANGPT2 antagonism of TIE2 causes destabilization of the endothelium in most vascular beds and promotes inflammation, vascular leakage, and impairment of the endothelial glycocalyx [25,26,27,28,29,30,31,32,33,34,35]. In addition, ANGPT2 can activate α5β1 integrin signaling [35]. Genetic- or antibody-mediated reduction of ANGPT2, as well as the administration of TIE2 activating agents, confer vascular protection and reduced mortality in experimental models of sepsis [12,22,36,37]. 

While the ANGPT2 inhibition of normal ANGPT1-TIE2 signaling probably has a role in endothelial dysfunction and coagulopathy, it was also shown in vitro that ANGPT2, and to a lower extent ANGPT1, can bind and inhibit the function of thrombomodulin (TM) [38]. TM is constitutively expressed on the luminal surface of endothelial cells where it is an important member of the intrinsic anticoagulant pathway, and an anti-inflammatory agent through the activation of the endothelial protein C receptor [39,40]. TM inhibits the procoagulant functions of thrombin by binding and inhibiting its interaction with procoagulant substrates and promoting instead the thrombin-catalyzed activation of protein C (APC) [39,41]. Endothelial-specific knockout of TM in mice disrupts the activation of protein C and causes lethal thrombus formation [42], highlighting the potency of this pathway. 

In the present study, we aimed at investigating the novel mechanism of ANGPT2 inhibition of TM in vivo, and if this could be an additional mechanism for hypercoagulation in critically ill COVID-19 patients. To examine this, we utilized in vivo models with different levels of ANGPT2 and evaluated the effect on the coagulation system with tail bleeding experiments. In addition, we performed TEG, and protein C activation assays on human plasma to study ANGPT2-mediated inhibition of TM in the absence of endothelial factors. Furthermore, we measured plasma concentrations of ANGPT2 and evaluated coagulation parameters in relation to hypercoagulation and clinical outcome in a cohort of critically ill COVID-19 patients. 

## 2. Materials and Methods

### 2.1. Study Design and Patients

The present study analyzed the plasma and clinical data of critically ill patients with COVID-19 included in the PronMed study, approved by the national Ethical Review Agency (EPM; No. 2020-01623). Between 13 March and 14 August 2020, 123 patients older than 18 years of age were included. All of the patients were diagnosed with COVID-19 by positive reverse-transcription PCR from nasopharyngeal swabs. In total, samples from 61 patients were included in the study, based on severity of disease and length of stay. Out of these, 40 patients with an extended length of stay (>10 days) were included with one sample at admission (day 1–4), and one sample at day 10–14. In addition, 21 patients with severe disease but a shorter length of stay were included, with one sample at admission (day 1–4). Apart from the standard ICU care and medications, all patients received thromboprophylaxis with dalteparin sodium at 100 IU/kg. Blood was sampled in 0.129 M trisodium citrate tubes (9NC BD Vacutainer, Becton Dickinson, Stockholm, Sweden), centrifugated at 2500× *g* for 10 min, and plasma was transferred and stored at −80 °C until analysis.

Clinical data were recorded prospectively, including medical history, medications, physiological data, and date of death. The Simplified Acute Physiology Score 3 (SAPS-3) [43], Sequential Organ Failure Assessment (SOFA) score [44], renal function “kidney disease improving global outcomes” (KDIGO) [45], circulatory support, and respiratory support data were collected, as detailed in the Results section.

Healthy control plasma was collected after consent from forty consecutive adult blood donors (43% females, 43 (37–49) years old) visiting the Blood Central at Uppsala University Hospital on 23 June 2020. TEG was performed on freshly collected plasma from seven donors (four females/three males, 30–50 years old). 

### 2.2. Blood Examinations

All routine lab tests were performed at the hospital’s clinical chemistry department. ELISA’s were used to measure the plasma protein concentrations for ANGPT2 (DANG20, R&D Systems, Minneapolis, MN, USA), ANGPT1 (DANG10, R&D Systems, Minneapolis, MN, USA), von Willebrand factor (ab108918, Abcam, Cambridge, UK), and ADAMTS13 (ab234559, Abcam, Cambridge, UK), according to the manufacturer’s instructions. Data for thrombin–antithrombin (TAT) complexes were available for some patients from a previous study [10]. 

### 2.3. Mice

Mice for the tail bleeding experiment came from in-house breeding on a C57BL/6J background (JAX stock no 000664). All experiments were performed in 8–20 week old female and male mice, as indicated below. Floxed *Angpt2* mice [46] and *Tie2* mice [46] were crossed to tamoxifen inducible *Cdh5*-Cre^ERT2^ [47] mice to generate endothelial specific knockout of *Angpt2* (*Angpt2*^iECKO^) and *Tie2* (*Tie2*^iECKO^). Wildtype (WT) controls were littermate mice with wt/wt alleles for *Angpt2* or *Tie2*, respectively, and positive for *Cdh5*-Cre^ERT2^. Mice were genotyped, as described previously [46], and knockout was induced with three doses of tamoxifen (2 mg) in peanut oil by oral gavage at 4 weeks of age. Wildtype control mice also received tamoxifen, as above.

### 2.4. Tail Bleeding Experiments

Mice with isoflurane anesthesia (~2% vol/vol) mixed with air (~0.5 L/min) were subjected to surgical dissection of the tail (3 mm from the tip) [48]. The tail was prewarmed for 2 min before dissection and immediately after immersed in buffered saline prewarmed to 37 °C. The time of bleeding was recorded. Experiments were performed in C57BL/6J mice 15 min or 2 h after receiving an i.p. injection of 8 nmol (low dose) or 80 nmol (high dose) recombinant human His-tagged fragments of ANGPT2 (ab220589, Abcam, Cambridge, UK), ANGPT1 (ab69492, Abcam, Cambridge, UK), or IgG (ab219660, Abcam, Cambridge, UK) in 100 µL PBS. In the low dose experiments, mice were 8–16 weeks old, and the number of mice used were *n* = 9 (5F/4M), *n* = 8 (5F/3M), and *n* = 9 (5F/4M) for ANGPT2, ANGPT1, and IgG injected mice, respectively. For the high dose experiments, mice were also 8–16 weeks old, and the number of mice used were *n* = 7 (4F/3M), *n* = 12 (10F/2M), and IgG (10F/2M), for ANGPT2, ANGPT1, and IgG, respectively. Tail bleeding experiments were also performed on *Angpt2*^iECKO^, *Tie2*^iECKO^, and littermate controls (WT). In the *Angpt2*^iECKO^ experiments, mice were 6–8 weeks old and *n* = 9 (6F/3M) and *n* = 7 (3F/4M) for WT and *Angpt2*^iECKO^, respectively. In the *Tie2*^iECKO^ experiments, mice were 18–20 weeks old and *n* = 5 (4F/1M) and *n* = 6 (4F/2M) for WT and *Tie2*^iECKO^, respectively.

Blood samples were collected by heart puncture in 1:10 in citate-dextrose anticoagulant (C3821, Sigma, St. Louis, MO, USA), centrifuged and prepared as above. Lungs were harvested, snap frozen, and stored at −80 °C for later protein analysis. One lobe of lung tissue was fixed in 4% PFA for 4 h, incubated overnight in 30% sucrose, embedded in optical cutting temperature medium, frozen, and stored at −80 °C. Plasma concentrations of injected recombinant human ANGPT2 were measured by ELISA, as above (DANG20, R&D Systems, Minneapolis, MN, USA). In addition, ELISA was performed on plasma for ANGPT2 (ab209883, Abcam, Cambridge, UK), activated protein C (abx052784, Abbexa, Cambridge, UK), and thrombin/antithrombin (TAT) complexes (ab137994, Abcam, Cambridge, UK), according to the manufacturer’s instructions. *Tie2* knockout was evaluated by a real-time PCR of kidney tissue with probes for *Tek* (*Tie2*) (Mm00443243_m1) and the housekeeping gene *Hprt* (Mm03024075_m1) from Thermo Fisher Scientific (Waltham, MA, USA).

### 2.5. ANGPT2 Association with Thrombomodulin In Vivo

Immunohistochemistry was performed on 14 µm thick lung cryosections after protein block (X0909, DAKO) with 0.25% Triton X100 for 1 h at room temperature. Sections were incubated over night at 4 °C with rabbit anti-mouse thrombomodulin (1:200, Ab230010, Abcam, Cambridge, UK) and goat anti-human ANGPT2 (1:200, AF623, R&D Systems, Minneapolis, MN, USA). Sections were washed with PBS containing 0.05% Tween-20. Sections were incubated with fluorescence-conjugated secondary antibodies for 2 h at room temperature, donkey anti-rabbit IgG (1:200, A32802, Thermo Fisher Scientific, Waltham, MA, USA) and donkey anti-goat IgG (1:200, A32816, Thermo Fisher Scientific, Waltham, MA, USA). Nuclei were stained with Hoechst 33342 (1:1000, P36930, Thermo Fisher Scientific, Waltham, MA, USA). Sections were washed as above, and mounted with ProLong Gold mounting media (H3570, Thermo Fisher Scientific, Waltham, MA, USA). Images were taken at 400× using a Leica SP8 confocal microscope.

Immunoprecipitation experiments were performed to evaluate the binding of ANGPT2 and ANGPT1 to thrombomodulin after injection into the mice. Total protein from the lung was extracted by homogenizing tissue in RIPA buffer (89901, Pierce, Thermo Fisher Scientific, Waltham, MA, USA) containing protease and phosphatase inhibitors (A32959, Pierce, Thermo Fisher Scientific, Waltham, MA, USA). After centrifugation, supernatant was collected and measured for protein concentration using a BCA assay (23227, Pierce, Thermo Fisher Scientific, Waltham, MA, USA), aliquoted and stored at −80 °C. Lung lysates (0.5 mg) were immunoprecipitated with a rabbit anti-mouse thrombomodulin antibody (ab230010, Abcam, Cambridge, UK) attached to protein G conjugated Dynabeads (10004D, Thermo Fisher Scientific, Waltham, MA, USA). Immunoprecipitated proteins were separated on 4–20% Mini Protean TGX gels (4561094, Biorad, Hercules, CA, USA) under reducing conditions and then transferred using Trans-blot turbo to 0.2 µm PVDF membranes (1704156, Biorad, Hercules, CA, USA). Blots were blocked with 5% BSA for 1 h and incubated overnight with mouse anti-6X His-tag antibody (27E8, 2366, Cell Signaling, Danvers, MA, USA). After washing and incubating with anti-mouse IgG-HRP-conjugated secondary antibody (NA931, Sigma, St. Louis, MO, USA), proteins were visualized using ECL plus detection reagents (GERPN2232, Sigma, St. Louis, MO, USA). Blots were stripped with Re-Blot Plus Strong solution (2504, Millipore, Burlington, MA, USA), blocked, and probed with anti-thrombomodulin antibody followed by anti-rabbit HRP conjugated secondary antibody (711-035-152, Jackson Immuno Research, Ely, UK). Western blotting of the lung lysates for thrombomodulin were visualized with the same antibodies as above, and HRP conjugated anti-GAPDH (ab9482, Abcam, Cambridge, UK) as the loading control. Band densities were measured in ImageJ (NIH).

### 2.6. Effect of ANGPT2 on TM Dependent Anticoagulation

Thromboelastography (TEG) on kaolin-activated plasma was utilized to evaluate the effect of ANGPT2 on thrombomodulin-dependent anticoagulation. The principles of TEG measurements were described previously [49]. Freshly drawn human blood, from both female and male donors, was sampled in 0.129 M trisodium citrate tubes (9NC BD Vacutainer, Becton Dickinson, Franklin Lakes, NJ, USA), and centrifugated at 2500× *g* for 10 min. Plasma was maintained at room temperature throughout the experiments. For each experiment, 1 mL of plasma was incubated in kaolin tubes (6300, Haemonetics, Boston, MA, USA) with 20 nM recombinant human soluble thrombomodulin (sTM) (3947-PA-010, R&D Systems, Minneapolis, MN, USA) and/or 40 nM ANGPT2 (ab220589, Abcam, Cambridge, UK) at room temperature for 20 min. The sTMs used in these experiments still have the domains for thrombin binding and anticoagulant activity [50]. TEG measurements were started by the addition of 340 µL kaolin plasma to TEG cups (6211, Haemonetics, Boston, MA, USA) containing 20 µL 0.2 M CaCl_2_ (7003, Haemonetics, Boston, MA, USA) in a TEG5000 (Nordic Biolabs). Analysis was continued for 15 min + reaction time (R), and samples were run in duplicate and averaged. From each donor, a duplicate sample without additives was run at the start and end of the experiment and averaged as the control.

### 2.7. Effect of ANGPT2 on Thrombomodulin Dependent Activation of Protein C

Pooled plasma from healthy controls was incubated with the same volume of 6 mM CaCl_2_ with 0.2 U/mL thrombin (T8885, Sigma, St. Louis, MO, USA), 20 nM of recombinant human sTM (3947-PA-010, R&D Systems, Minneapolis, MN, USA), and 40 nM ANGPT2 (ab220589, Abcam, Cambridge, UK), ANGPT1 (ab69492, Abcam, Cambridge, UK) or IgG (ab219660, Abcam, Cambridge, UK) at 37 °C for 30 min. The reaction was terminated by addition of 0.2 U/mL hirudin (H0393, Sigma, St. Louis, MO, USA) at 37 °C for 10 min. After addition of 50 µL chromogenic APC substrate (229021, Biophen CS-21(66)) the increase in absorbance was measured at 405 nm for 8 min (linear phase) at 37 °C in a plate reader (Synergy HT, Biotek, Winooski, VT, USA). The area under the curve was used to calculate APC concentrations expressed as arbitrary units (a.u.).

### 2.8. Study Approval

The study was approved by the Swedish National Ethical Review Agency (EPM; No. 2020-01623). Informed consent was obtained from the patient, or next of kin if the patient was unable to give consent. Healthy blood donor samples were approved under ethical permit No. 01/367. The Declaration of Helsinki and its subsequent revisions were followed. The protocol for the study was registered (ClinicalTrials ID: NCT04316884); STROBE guidelines were followed for reporting clinical data and ARRIVE 2.0 for animal data. All animal experiments were approved by the Uppsala Committee of Ethics of Animal Experiments (approved permit numbers 5.8.18-04862-2020 and 5.8.18-03858-2021) and were conducted according to guidelines established by the Swedish Board of Agriculture.

### 2.9. Statistics

Data are expressed as geometric mean ± 95% confidence interval (CI). To test for statistical differences, we utilized the Student’s *t*-test or one-way ANOVA (>2 groups) where appropriate. ANOVA was followed by Bonferroni’s post hoc test. Data were tested for normality and unevenly distributed data were log transformed before statistical analysis. Pearson’s correlation was used to measure dependence between two variables, except for the SAPS-3, SOFA score, and TEG MA, where non-parametric Spearman’s correlation was used. Fisher’s exact test was used to evaluate patients’ results in comparison to a given reference interval representing the normal range based on mean ± 2 SD (as calculated in the clinic). Kaplan–Meier plots with Log-rank test were used to estimate the probability of survival. The optimal cut-off value was determined from Receiver Operating Characteristics (ROC) curve analysis, and the cut-off with the highest likelihood ratio was selected i.e., for ANGPT2, the area under the ROC curve *p* = 0.0004; cut-off 8.3 ng/mL with a likelihood ratio of 23.5. All of the statistical analysis was done in GraphPad Prism 9. All analyses were two-sided and a *p* < 0.05 was considered significant.

## 3. Results

### 3.1. ANGPT2 Administration Decreases Tail Bleeding Time in Mice

To elucidate the effect of ANGPT2 on the coagulation system in vivo, we utilized tail bleeding experiments. Tail bleeding experiments in rodents are commonly used to evaluate factors that affect the coagulation system [48], and previous administration of sTM was shown to increase bleeding times [51]. Recombinant His-tagged human ANGPT2, ANGPT1, or IgG, were injected before bleeding experiments. ANGPT1 was included to compare with ANGPT2 in vivo, as in vitro studies show that ANGPT1 also bind TM, albeit with lower affinity than ANGPT2 and preferentially interacting with TIE2 in a competitive situation [38]. Incubation times were kept short to avoid transcriptional changes of the endothelium. Intraperitoneally injection of 8 nmol human ANGPT2 for 2 h resulted in a 50% reduction (*p* < 0.001) in tail bleeding time, while mice with ANGPT1 or IgG injections were unaffected (Figure 1a). Importantly, ANGPT2 administration resulted in circulating ANGPT2 levels equivalent to that of plasma from COVID-19 patients (Figure 1b). There was no added effect of increasing the dose 10-fold (80 nmol) as a similar reduction in bleeding time was seen, but as expected, circulating ANGPT2 increased 10-fold (Figure 1b). The higher dose showed an effect already 15 min after injection, while the low dose needed longer circulation time to affect tail bleeding time. The in vivo inhibition of TM by ANGPT2 was supported by a significantly reduced plasma level of activated protein C (APC) (Figure 1c). Thrombin/antithrombin (TAT) complexes reflect circulating levels of thrombin as thrombin is quickly bound to antithrombin. ANGPT2 administration resulted in a 3-fold increase (*p* = 0.001) of TAT complexes in mice injected with ANGPT2 (Figure 1d). 

Immunohistochemical staining of lung tissue from ANGPT2 injected mice showed colocalization of thrombomodulin and human ANGPT2 (Figure 1e). IgG and ANGPT1 injected mice showed weak staining for Angpt2 as the antibody cross-reacts with mouse Angpt2. As the Tie2 expression pattern is similar to TM, immunohistochemistry could not determine specific ANGPT2/TM binding. To examine if ANGPT2/TM complexes were formed in our experiments, we performed immunoprecipitation experiments on lung tissue. Immunoprecipitation of TM showed positive blotting for His-tag in mice injected with ANGPT2 (Figure 1f,g; Appendix A), confirming the presence of ANGPT2/TM complexes. Immunoprecipitation experiments from the lung suggested a reduction of TM in the tissue after the ANGPT2 injection. Western blotting for TM in the lung lysates showed a significant decrease of TM after ANGPT2 injection (Figure 1h,i; Appendix A). At the same time, the sTM in plasma increased significantly (*p* < 0.01) (Figure 1j).

### 3.2. Endothelial ANGPT2 Deficiency Increases Tail Bleeding Time

As circulating ANGPT2 levels are low in healthy people, we next wondered if ANGPT2 could also influence the coagulation system in normal physiology. To investigate this, we utilized mice with an inducible knockout of Angpt2 specifically in endothelial cells (Angpt2^iECKO^), which had significant (*p* = 0.0003) decreased circulating ANGPT2 (Figure 2a). In tail bleeding experiments, Angpt2^iECKO^ mice had a 2.5-fold (*p* < 0.001) increase in bleeding time (Figure 2b). This experiment supports an ANGPT2-mediated regulation of the coagulation system in normal physiology. ANGPT2 is also known to inhibit TIE2 signaling in inflammatory situations. Previously, heterozygous Tie2 mice showed an increased thrombus formation after laser injury [22], while mice deficient for the Tie2 agonist Angpt1 had no difference in tail bleeding time [38]. To investigate if Tie2 deficiency alone could recapitulate the decrease in tail bleeding seen after ANGPT2 injection, we performed experiments in mice with induced endothelial specific knockout of Tie2 (Tie2^iECKO^). Tie2^iECKO^ mice had a 70% reduction of Tie2 (Appendix A). No difference in the tail bleeding experiments was observed between the wildtype mice and Tie2^iECKO^ mice (Figure 1c), supporting a Tie2 independent pathway in our experiments.

### 3.3. ANGPT2 Inhibits TM Mediated Anticoagulation in Human Plasma

To further study the effects of ANGPT2 on the coagulation system, we utilized thromboelastography (TEG) on human plasma to test the efficacy of blood coagulation in the absence of the endothelial compartment. TEG was performed on freshly collected plasma, at baseline, and after the addition of sTM, ANGPT2, or both. The recombinant sTM contains the domains necessary for thrombin binding and anticoagulant activity [50]. TEG curves from the different conditions from one of the high responder donors can be seen in Figure 3a, with measured parameters indicated. As expected through its anticoagulative properties, sTM significantly (*p* < 0.001) increased the time for the clot formation to start (TEG R, reaction time) (Figure 3b). For some donors, an effect could also be seen for sTM on thrombus strength (TEG MA, maximal amplitude), however, this did not reach statistical significance (Figure 3c). The addition of ANGPT2 to the TM-supplemented plasma completely blocked the sTM-mediated anticoagulation, and, importantly, ANGPT2 alone did not affect the TEG parameters (Figure 2b,c). Results from the in vitro TEG experiments are summarized in Table 1. These experiments support the notion that ANGPT2 can inhibit TM-mediated anticoagulation in the absence of the endothelial compartment, thus, independently of TIE2 inhibition or α5β1 integrin activation.

### 3.4. ANGPT2 Inhibits TM Mediated Activation of Protein C

TM inhibits the procoagulant functions of thrombin by binding and inhibiting its interaction with procoagulant substrates and instead promoting thrombin-catalyzed activation of protein C [39,41]. The activation of protein C is a direct effect of thrombin/TM function and can be measured by a chromogenic substrate in plasma after activation with thrombin. We utilized human plasma from healthy donors expected to have protein C but low endogenous sTM levels. The plasma was incubated with recombinant sTM, ANGPT2, ANGPT1, and IgG, either alone or in combination to evaluate their influence on thrombin/TM-mediated activation of protein C. As expected, sTM significantly (*p* < 0.001) increased the activation of protein C (Figure 4). The addition of ANGPT2 significantly reduced (*p* < 0.001) activation of protein C (Figure 4). In contrast to the in vivo situation, a significant inhibition of TM function could also be seen with the addition of ANGPT1. This is probably due to a lack of competitive binding to TIE2, which ANGPT1 would bind with higher affinity compared to TM [38], please see the Discussion section below.

### 3.5. COVID-19 Patient Cohort

The study included a cohort of 61 patients admitted to the ICU at Uppsala University Hospital with COVID-19. Plasma samples were collected 1–4 days (61 patients) and 10–14 days after admission for patients remaining in the ICU > 10 days (40 patients). Patient demographic characteristics are summarized in Table 2. All of the patients received prophylactic anticoagulation therapy with dalteparin sodium during the ICU stay [4], steroids were given to seven patients (11%), and one patient received remdesivir. Most of the selected ICU patients received invasive mechanical ventilation (80%), and 25 patients died (41%).

### 3.6. ANGPT2 Is Elevated in Critically Ill COVID-19 Patients and Correlates with Disease Severity

Earlier studies showed that ANGPT2 is elevated in COVID-19 patients and correlates to disease severity [12,15,19,23,24]. In the present study, we measured ANGPT2 and other parameters in plasma at two different timepoints (admission, and after 10 days) to evaluate changes over time in a selected cohort of 61 critically ill COVID-19 patients. ANGPT2 was elevated in all of the patients at ICU admission and continued to increase over time in the ICU (Table 3). Further analysis showed that non-recovering patients had 42–55% (*p* < 0.001) higher ANGPT2 levels compared to recovering patients at both timepoints (Figure 5a). Kaplan–Meier survival plots utilizing a cut-off value of 8.3 ng/mL (see the Statistics section for further details) demonstrated that the ANGPT2 levels were strongly predictive (*p* < 0.0001) of survival in the studied cohort (Figure 5b). Plasma concentrations of the TIE2 agonist ANGPT1 were minimally affected (Figure 5c). Other clinical and measured parameters of the patients during their ICU stay are summarized in Table 3. 

Further analysis showed that ANGPT2 correlated with several markers of disease severity (Figure 5d). The simplified acute physiology (SAPS-3) score (predicting the probability of death) correlated significantly (*p* < 0.05) with ANGPT2 at ICU admission. The sequential organ failure assessment (SOFA) score (extent of organ failure) was calculated for the same days as the collection of plasma samples. The SOFA score represents six organ systems, where each organ system is assigned a point value from 0 (normal) to 4 (high degree of dysfunction/failure) [44]. ANGPT2 and SOFA scores were significantly correlated (*p* < 0.01) (Figure 5d). In line with this, we found that COVID-19 patients with acute kidney injury (AKI) had higher ANGPT2 levels (*p* = 0.044) (Figure 5e), and that the ANGPT2 inversely correlated with eGFR (creatinine) (*p* < 0.0001) (Figure 5d). In contrast, we found no association of ANGPT2 with troponin I (myocardial injury) or pulmonary function (PaO2/FiO2) (Figure 5d). The inflammatory markers CRP, TNFa, IL6, and ferritin were all elevated in COVID-19 patients at admission (Table 3), and TNFa showed significant correlation with ANGPT2 (*p* < 0.01) (Figure 5d). Other inflammatory markers did not correlate with ANGPT2, and neither did lactate. These data show that ANGPT2 correlates with severity of disease, kidney function, and mortality.

### 3.7. Hypercoagulation in Critically Ill COVID-19 Patients

To investigate if ANGPT2 correlated with hypercoagulation in these patients, we assessed several markers of the coagulation system including TAT complexes, D-dimer, fibrinogen, thromboelastography (TEG) parameters, platelets, Von Willebrand factor (VWF), and ADAMTS13, which are summarized in Table 3. TAT complexes were significantly (*p* < 0.001) increased in all of the patients and correlated (*p* < 0.01) with ANGPT2 (Figure 5d, Table 3). D-dimer and fibrinogen levels had a similar pattern and were increased in all of the patients and showed significant correlation (*p* < 0.05) with ANGPT2 (Figure 5d, Table 3). Platelet counts were increased in patients at the late timepoint (but still within normal range) (Table 3). Patients with TEG MA values > 69 mm had over 60% (*p* < 0.05) higher ANGPT2 levels than patients with <69 mm (Figure 5f). TEG MA increased over time in the ICU and correlated significantly (*p* < 0.05) with ANGPT2 (Figure 5d). Other TEG parameters are presented in Table 3. These data show that ANGPT2 correlates to several markers of hypercoagulation in critically ill COVID-19 patients.

Von Willebrand factor (VWF) is stored and released from platelets and endothelial Wiebel–Palade bodies [52]. As VWF and ANGPT2 were reported to be stored in the same endothelial vesicles [53], we assessed the plasma levels of VWF. VWF was significantly (*p* < 0.0001) elevated in COVID-19 patients compared to healthy controls and correlated (*p* < 0.05) to ANGPT2 (Table 3, Figure 5d, Appendix A). ADAMTS13 is a metalloprotease produced by the liver that degrades large VWF multimers, thereby decreasing VWF’s pro-coagulation properties [54]. We found that ADAMTS13 was significantly (*p* < 0.003) decreased at the late timepoint, especially in non-recovering patients, compared to all of the other patients, suggesting an increased consumption of ADAMTS13 (Table 3, Figure 5d, Appendix A). VWF correlated with the ANGPT2, SOFA score, eGFR (creatinine), P-creatinine, platelet count, and TEG MA, but not with SAPS-3. ADAMTS13 correlated with SAPS-3 and TNFa. Neither VWF nor ADAMTS13 could predict mortality in the cohort.

**Figure 5 biomedicines-10-01333-f005:**
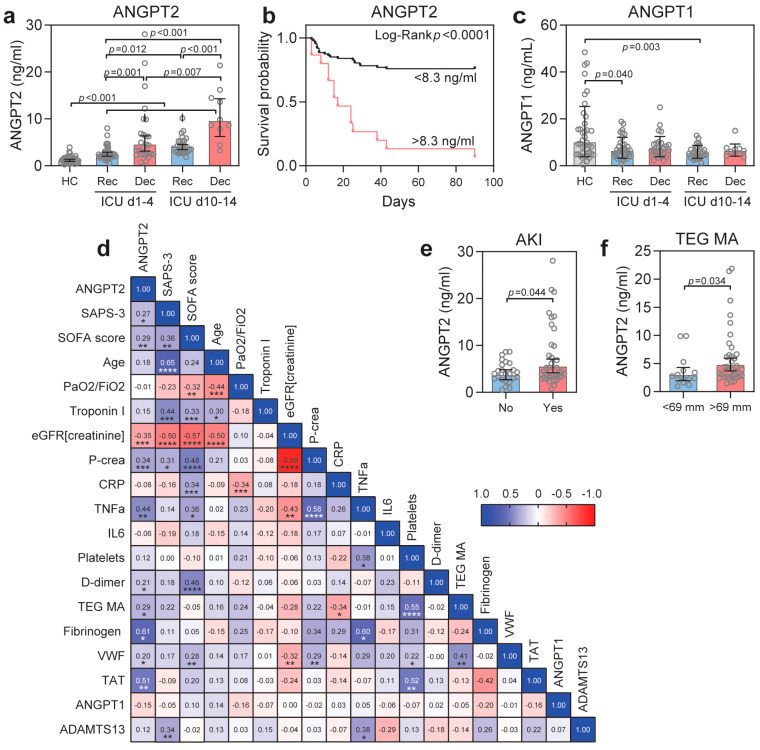
(**a**) Plasma ANGPT2 concentrations for healthy controls (HC), recovered (Rec), and deceased (Dec) patients at day 1–4 and day 10–14 after admission; (**b**) Kaplan–Meier survival plot with Log-rank test show that ANGPT2 levels can predict survival; (**c**) Plasma ANGPT1 was minimally affected; (**d**) Clinical data and measured parameters were analyzed by Pearson correlation test (Spearman for SAPS-3 and SOFA score) and presented in a heatmap for correlation (blue) or inverse correlation (red), * *p* < 0.05, ** *p* < 0.01, *** *p* < 0.001, **** *p* < 0.0001; (**e**) ANGPT2 is higher in patients with acute kidney injury (AKI) compared to non-AKI patients; (**f**) ANGPT2 is higher in patients with TEG MA > 69 mm. Data are presented as mean ± 95% CI. Statistical comparison from one-way ANOVA with Bonferroni post hoc test (**a**,**b**) or Student’s *t*-test (**e**,**f**). AKI, acute kidney injury; TEG, thromboelastography; MA, maximal amplitude.

**Table 3 biomedicines-10-01333-t003:** Clinical parameters of patients during ICU stay.

	Day 1–4	*n*	Day 10–14	*n*	*p*-Value *	Reference Range	*p*-Value †
ANGPT2 (ng/mL)	3.1 (2.6–3.8)	61	4.8 (4.0–5.8)	40	*p* = 0.003	1.2 (1.0–1.4) ^‡^	*p* < 0.001
ANGPT1 (ng/mL)	6.2 (4.9–7.8)	61	5.2 (4.4–6.3)	40	ns	9.9 (7.3–13.4) ^‡^	ns
VWF (IU/mL)	3.4 (3.2–2.7)	61	3.6 (3.2–4.0)	40	ns	1.5 (1.4–1,7) ^‡^	*p* < 0.001
ADAMTS13 (ng/mL)	887 (760–1035)	61	640 (490–837)	40	*p* = 0.003	930 (817–1059) ^‡^	ns
Platelets (×10^9^/L)	231 (208–258)	58	426 (353–514)	37	*p* < 0.001	150–300	ns
D-dimer (mg/L)	2.27 (1.70–3.03)	60	2.49 (1.85–3.35)	34	ns	<0.50	*p* < 0.001
CRP (mg/L)	192 (164–224)	59	65 (45–95)	38	*p* < 0.001	<5	*p* < 0.001
SOFA score	6 (6–7)	60	6 (5–7)	40	ns	0	*p* < 0.001
PaO_2_/FiO_2_	18.6 (17.0–20.3)	57	23.2 (20.9–25.7)	37	*p* = 0.002	>60	*p* < 0.001
Ferritin (µg/L)	1377 (1033–1837)	56	1417 (979–2068)	36	ns	25–310	*p* < 0.001
Lactate (mmol/L)	1.3 (1.2–1.4)	59	1.2 (1.1–1.4)	38	ns	0.8–2.0	ns
TNFa (ng/L)	66.7 (56.4–79.0)	40				<8.0	*p* < 0.001
IL6 (ng/L)	156 (120–204)	35				<7.0	*p* < 0.001
Fibrinogen (g/L)	8.4 (7.6–9.3)	15				2.0–4.2	*p* < 0.001
TAT (µg/l)	49 (37–65)	32				<10	*p* < 0.001
**TEG**							
R (min)	8.7 (7.5–10.0)	30	12.3 (8.6–17.5)	22	ns	4.6–9.1	ns
R_KCH_ (min)	7.3 (6.8–8.0)	30	9.2 (8.0–10.6)	22	*p* = 0.003	4.3–8.3	ns
Angle (deg)	77.1 (75.8–78.4)	30	76.5 (74.2–78.9)	22	ns	64–77	ns
MA (mm)	68.8 (67.2–70.4)	30	73.6 (72.1–75.1)	22	*p* < 0.001	52–69	ns
LY30 (%)	0.1 (0–0.2)	30	0.1 (−0.1–0.2)	22	ns	0–2.6	ns

Data are expressed as mean ± 95% CI. TAT, Thrombin/Antithrombin complex; CRP, C-reactive protein; PaO_2_/FiO_2_, the ratio of arterial oxygen partial pressure (PaO_2_ in mmHg) to fractional inspired oxygen; TEG, thromboelastography; R, reaction time; KCH, kaolin with heparinase; MA, maximum amplitude. * *p*-values from unpaired *t*-test comparing patients at day 1–4 and patients at day 10–14. † *p*-values from Fisher’s exact test comparing patients at day 1–4 with reference range. ‡ measured in healthy controls in the study.

## 4. Discussion

To test the hypothesis that ANGPT2 binds and inhibits TM function in vivo, we investigated the effect of ANGPT2 administration in mice on tail bleeding time. ANGPT2 administration decreased tail bleeding time and increased circulating TAT complex, supporting a shift to a procoagulant state. The presence of ANGPT2/TM interaction was confirmed in lung tissue after ANGPT2 injection. In contrast, Angpt2 deficient mice showed increased tail bleeding times, suggesting an intricate role for ANGPT2 in balancing TM function in normal physiology. To investigate the role for ANGPT2 in COVID-19-mediated coagulopathy, we studied plasma from critically ill COVID-19 patients at two time points, together with clinical features. Our data show that circulating ANGPT2 levels are increased in critically ill COVID-19 patients, and that ANGPT2 continues to increase over time in the ICU. Importantly, ANGPT2 levels correlated with disease severity (SAPS-3, SOFA, score), hypercoagulation (TAT complex, D-dimer, TEG, fibrinogen, VWF), and mortality. 

In the current study, we show for the first time that ANGPT2 can bind and regulate TM function in vivo (Figure 6). We utilized the tail bleeding model to evaluate the coagulation system after ANGPT2 administration, as this model is well-established and previously showed sensitivity to sTM administration [51]. Experiments were done with short exposure to ANGPT2 (≤2 h) to minimize transcriptional changes that could influence the coagulation system. Ideally, we would have liked to evaluate APC in both mice and patients as a more direct measure of TM function, but this proved to be technically difficult in patient plasma samples. Instead, we measured the circulating TAT complexes as these reflect intravascular thrombin generation. TAT complexes were previously shown to change with thrombomodulin activity, TAT increased in thrombomodulin-deficient mice [55], and decreased in primates after thrombomodulin administration [56]. 

Because of the suggested co-storage of ANGPT2 and VWF in endothelial Weibel–Palade bodies [53], we investigated VWF, and ADAMTS13, a liver-derived plasma metalloprotease that degrades large VWF multimers and thereby decreases VWF’s pro-hemostatic properties [54]. VWF was significantly increased in all of the ICU patients compared to healthy controls, but, in contrast to ANGPT2, we found no differences in VWF levels between the ICU patient groups. This contrasts with a recent study reporting a correlation between VWF and mortality in ICU COVID-19 patients [18]. Previously, decreased concentrations of ADAMTS13 were shown to correlate with mortality in COVID-19 and septic shock patients [19,57]. It is plausible that this progressive decrease reflects increased consumption, rather than lowered production, of ADAMTS13. In agreement with this, the ADAMTS13 levels decreased significantly over time in the ICU in non-recovering patients in our cohort. We were unable to predict mortality in our cohort by ADAMTS13, however, this may be to the limited number of non-recovering patients at the late time point. 

ANGPT2 is stored in endothelial secretory vesicles and exocytosis can be stimulated by inflammatory cytokines including thrombin, histamine, and TNFa [30,58]. Until now, most focus has been on ANGPT2′s antagonistic effect on TIE2 signaling in disease. There is no doubt that TIE2 signaling has a critical role in preventing vessel permeability and presenting an anti-thrombotic surface by remaining activated via continuous secretion of ANGPT1 from perivascular cells and platelets [59]. Importantly, TIE2 signaling limits ANGPT2 transcription [22,60,61], preserves endothelial integrity and glycocalyx important for antithrombin activity [62,63], and maintains production of endothelial-derived nitric oxide that may decrease platelet aggregation [64,65]. TIE2 activating strategies are known to improve outcomes in models of disease, including sepsis [22,36,37]. Schmaier et al. recently showed that endothelial cells in vitro exposed to plasma from critically ill COVID-19 patients induced thrombo-inflammatory gene expression and activation of coagulation, including ANGPT2 expression and generation of factor Xa and thrombin [12]. The endothelial cell transcription of many of these factors, including ANGPT2, PROCR, and THBD, were normalized upon stimulation of TIE2 by recombinant ANGPT1 or a small molecule inhibitor of VEPTP (decreasing dephosphorylation of TIE2) [12]. In the current study, we did not observe changes in tail bleeding time in *Tie2* deficient mice, and neither did Daly et al. in *Angpt1* deficient mice [38]. This contrasts with a previous study that showed increased thrombosis in heterozygous Tie2 mice after laser injury [22]. Our studies used different methods and vascular beds which may be an explanation for the variability. Currently, we have no means of assessing the local concentrations of ANGPT2 in vivo. Endothelial expression of ANGPT2 varies extensively among vessel types and location and is regulated in response to angiogenic and inflammatory activation. TM and TIE2 are highly expressed in all of the endothelial cells along the arterial-venous zonation, at least in the lung [66]. It is therefore possible that inhibition of TM or TIE2 occurs only locally at sites of high ANGPT2 release and that this may differ between vascular beds and organs.

While both ANGPT2 and ANGPT1 can bind both TM and TIE2, the dynamic is different. ANGPT2 has a higher binding affinity to TM compared to ANGPT1. Daly et al. demonstrated that ANGPT2 only required a concentration of ~8.3 nM to saturate TM binding, while the required concentration for ANGPT1 was ~88 nM [38]. Furthermore, in a competitive situation with both receptors, ANGPT1 bound almost exclusively to TIE2 in contrast to ANGPT2 that bound to both receptors. This is to some extent visualized in our in vivo experiments with both receptors present, where ANGPT1 administration does not change tail bleeding time, activation of protein C, or alter TAT complex formation. Our in vitro studies show that when only sTM is present, ANGPT1 can inhibit TM-mediated activation of protein C. It has also been suggested that APC binds and activates TIE2 more directly [67], which needs further investigation. 

It should be noted that we did not perform any dose-response studies in our in vitro experiments for ANGPT2, and that the dose is above the measured circulating levels in the current study and literature. Daly et al. noted in their study that fairly high levels of ANGPT1 and ANGPT2 were needed to significantly inhibit TM in endothelial cell cultures [38]. Furthermore, we utilized sTM instead of the full-length version. The reason for this was the commercially availability of sTM compared to TM. The sTM used in the current study had the domains for thrombin binding and anticoagulative activity [50]. It has been reported that isolated sTM from human plasma has lower anticoagulative properties (30–50%) compared to endothelial full length thrombomodulin [68,69]. The reason for this is not known, but it is possible that the activity is lower per se, or that an inhibitor is bound (such as ANGPT2). 

Our results for ANGPT2 in COVID-19 patients are in line with recently published data [12,15,19,23]. The strength of the current study was the analysis of two different timepoints, making it possible to observe changes in the ANGPT2 and other parameters over time. We find that the ANGPT2 levels correlate with several markers of hypercoagulation and mortality. Interestingly, there was a strong inverse correlation between ANGPT2 and kidney function. Our data suggest that ANGPT2 levels are not only a biomarker for disease severity, but also causal and contribute to hypercoagulation in critically ill COVID-19 patients. ANGPT2 is increased in several diseases with coagulopathy, including sepsis, and it is plausible that the ANGPT2 inhibition of TM has a role there as well. 

Our data suggest that inhibition of ANGPT2 inhibition may be explored as a therapeutic approach in COVID-19 and other diseases with hypercoagulation. Trebananib (formerly AMG386) binds to both ANGPT2 and ANGPT1. While Trebananib may be an unsuitable ANGPT2 inhibitor, due to its ANGPT1 binding and inhibition of TIE2 signaling [37], there were not, to our knowledge, any adverse effects reported regarding the coagulation system [70,71]. There are currently many clinical trials registered with Angiopoietin-2 antibodies, several for the treatment of solid tumors. Critically ill COVID-19 patients may benefit from these antibodies. Another very interesting compound at the preclinical stage is ABTAA, a humanized Angiopoietin-2 Binding and TIE2-Activating Antibody [37]. ABTAA binds and clusters ANGPT2, converting it into a TIE2-activating molecule, while simultaneously decreasing the circulating ANGPT2. ABTAA treatment showed promising results in experimental models of sepsis; glycocalyx was restored, but the coagulation system was not evaluated [37]. It should be noted that ABTAA showed greater protection in experimental sepsis, compared to ANGPT2 inhibition alone [37]. 

Another approach would be to interfere with exocytosis and translation of ANGPT2. We found a significant correlation between ANGPT2 and TNFa in the current study, and patients may benefit from anti-TNFa treatments [72], as TNFa can induce exocytosis of ANGPT2, at least in vitro [58]. Anti-TNFa therapy in patients with rheumatoid arthritis decreased cytokines and acute phase proteins including IL-6, IL-1 receptor antagonist, serum amyloid A, haptoglobin, and fibrinogen [73,74,75]. Significant reductions could also be seen for D-dimer and pro-thrombin fragments with anti-TNFa therapy [76]. 

## 5. Conclusions

In conclusion, we show that the ANGPT2 levels in critically ill COVID-19 patients correlate with severity of disease, hypercoagulation, and mortality. In addition, we provide novel in vivo evidence for a direct role for ANGPT2 in the coagulation system through the binding and inhibition of TM-mediated anticoagulation. These findings suggest that inhibition of ANGPT2 might not only benefit critically ill COVID-19 patients but also other patients with hypercoagulation.

## Figures and Tables

**Figure 1 biomedicines-10-01333-f001:**
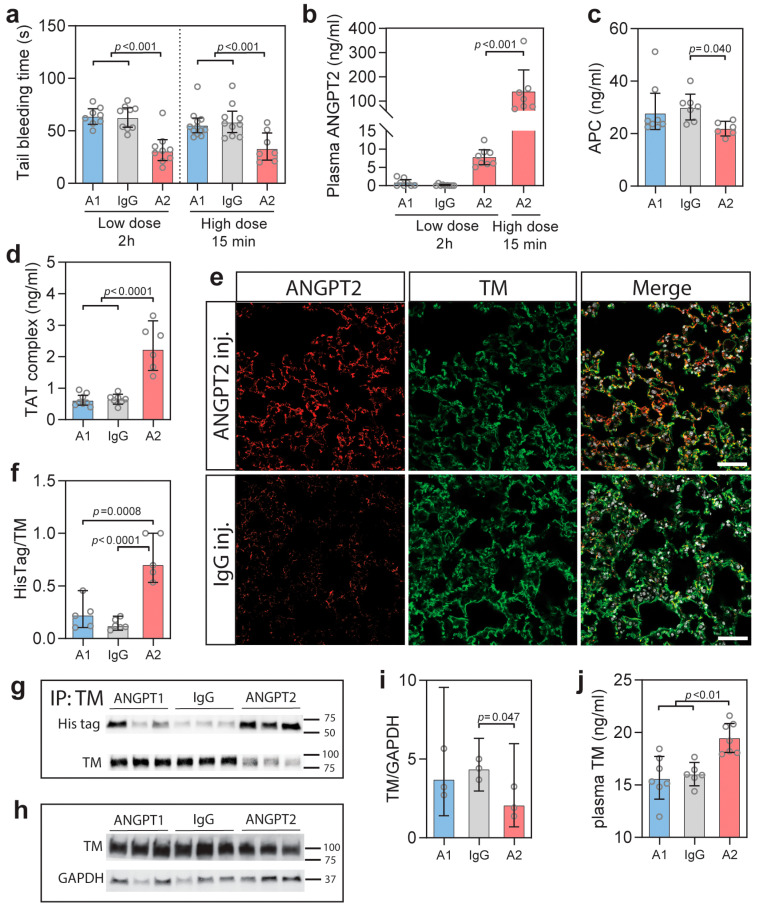
(**a**) Tail bleeding time after intraperitoneal (i.p.) injection of His-tagged ANGPT2 (A2), ANGPT1 (A1), and IgG with (8 nmol) and high dose (80 nmol) after 15 min or 2 h, as indicated; (**b**) Plasma concentrations of ANGPT2 after i.p. injections; (**c**) Plasma concentrations of activated protein C (APC) 15 after high dose injection of indicated proteins; (**d**) Plasma concentrations of thrombin/antithrombin (TAT) complexes 15 min after high dose injection of indicated proteins; (**e**) Immunohistochemistry for human/mouse ANGPT2 and TM on lung tissue after administration of high dose of ANGPT2 or IgG for 15 min; (**f**,**g**) Representative blot and quantification of His-tag labelled proteins after immunoprecipitation of TM in lung tissue 15 min after intraperitoneal (i.p.) injection of 80 nmol (high dose) of His-tagged ANGPT2 (A2), ANGPT1 (A1), and IgG. Full gels can be seen in the Appendix A; (**h**,**i**) Western blotting and quantification of TM in lung lysates from the same conditions as above. Full gels can be seen in the Appendix A; (**j**) Plasma sTM 15 min after injection of 80 nmol ANGPT2 (A2), ANGPT1 (A1), and IgG. Data presented as mean ± 95% CI. Statistical comparison from one-way ANOVA with Bonferroni post hoc test.

**Figure 2 biomedicines-10-01333-f002:**
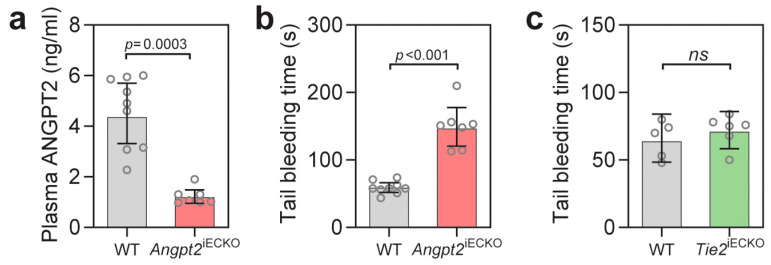
(**a**) Plasma concentration of ANGPT2 in Angpt2^iECKO^ mice and wildtype (WT) littermate controls; (**b**) Tail bleeding time in Angpt2^iECKO^ mice and wildtype (WT) littermate controls. (**c**) Tail bleeding time in Tie2^iECKO^ mice and littermate controls (WT). Data presented as mean ± 95% CI. Statistical comparison with Student’s *t*-test.

**Figure 3 biomedicines-10-01333-f003:**
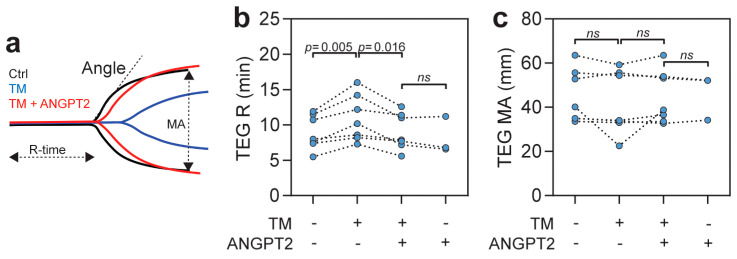
(**a**) TEG data from one of the donors showing sTM-dependent (blue) increase in reaction time (R) and decreased maximal amplitude (MA), which is inhibited by ANGPT2 (red); (**b**,**c**) TEG analysis of individual donor blood with 20 nM sTM) and 40 nM ANGPT2. Statistical comparison with repeated measures ANOVA with Bonferroni post hoc test. Additional TEG results can be found in Table 1.

**Figure 4 biomedicines-10-01333-f004:**
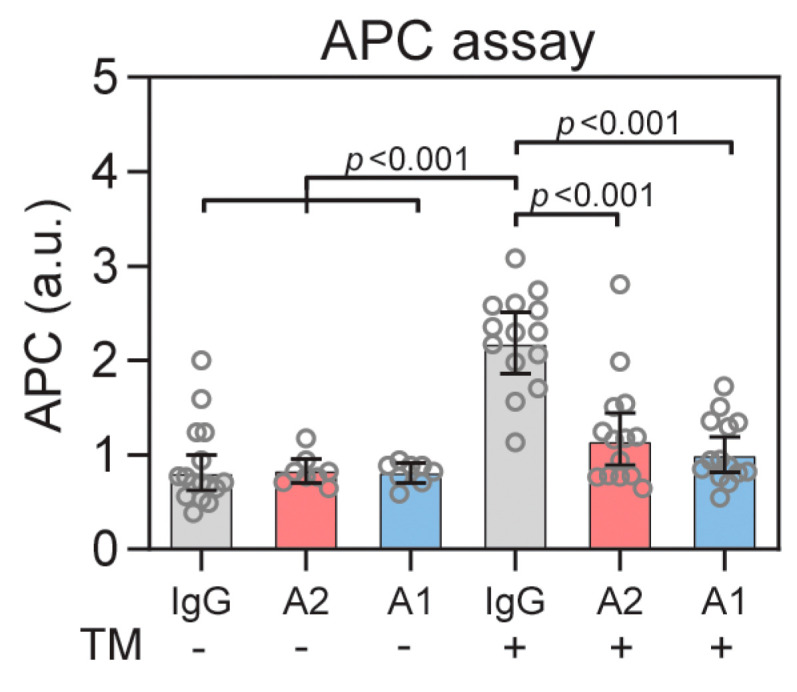
TM-dependent activation of protein C (APC) can be inhibited with ANGPT2 or ANGPT1. Human healthy donor plasma was supplemented with sTM (20 nM) and incubated with 40 nM ANGPT2, IgG or ANGPT1. Data are presented as mean ± 95% CI. Statistical comparison from one-way ANOVA with Bonferroni post hoc test.

**Figure 6 biomedicines-10-01333-f006:**
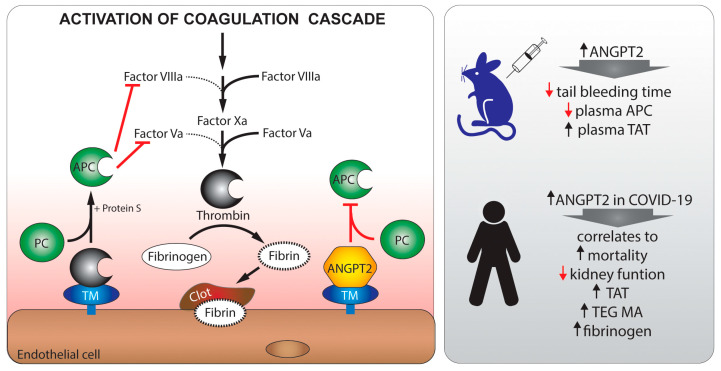
A schematic overview of ANGPT2 signaling in health and in hypercoagulation with high levels of ANGPT2 inhibiting TM-mediated anticoagulation. Thrombin/TM normally catalyzes protein C (PC) to activated protein C (APC), which inhibits Factor VIIIa and Factor Va. Increased levels of ANGPT2 in mice reduces tail bleeding time and plasma APC, while increasing TAT. In critically ill COVID-19 patients, ANGPT2 is increased and correlates to reduced kidney function, hypercoagulation, and mortality.

**Table 1 biomedicines-10-01333-t001:** TEG data from in vitro experiments.

	Control	sTM	sTM + ANGPT2	ANGPT2
R (min)	8.6 (6.6–11.2)	10.5 (8.0–13.8) ^a,b^	8.7 (6.7–11.4)	8.0 (3.8–16.6)
Angle (deg)	64.8 (69.6–70.2)	54.2 (41.8–70.1)	60.0 (51.1–70.3)	67.8 (59.4–77.4)
MA (mm)	45.5 (34.4–60.2)	40.7 (27.1–61.0)	45.3 (34.6–59.4)	45.3 (24.8–82.8)

Data are presented as mean ± 95% CI. R, reaction time; Angle, rate of clot formation; MA, maximal amplitude, sTM, soluble thrombomodulin, ANGPT2, Angiopoietin-2. Statistical comparison was with repeated measures one-way ANOVA. ^a^
*p* = 0.0051 vs. Control, ^b^
*p* = 0.0155 vs. sTM + ANGPT2.

**Table 2 biomedicines-10-01333-t002:** Patient demographic characteristic and comorbidities.

	All Patients (*n* = 61)
Women, *n* (%)	12 (20)
Age, years	60.9 (56.7–65.4)
Body weight, kg	88.6 (83.5–93.9)
BMI, kg/m^2^	30.1 (28.5–31.8)
COVID-19 day on ICU arrival	9.2 (8.3–10.3)
Invasive ventilation	49 (80)
90-day mortality	25 (41)
**Comorbidities, *n* (%)**	
Pulmonary disease	18 (30)
Hypertension	38 (62)
Heart failure	4 (7)
Peripheral vessel disease	17 (28)
Previous thromboembolic event	8 (13)
Diabetes	21 (34)
Malignancy	6 (10)
≥2 comorbidities	32 (26)
**Clinical features at arrival**	
ARDS, *n* (%)	20 (100)
SAPS-3	53 (51–56)

Data are expressed as *n* (%) or mean ± 95% CI. BMI, body mass index; ICU, intensive care unit; ARDS, acute respiratory distress syndrome; SAPS-3, simplified acute physiology score.

## Data Availability

The authors declare that all supporting data are available within the article and its online supplementary files. Individual level patient data can be made available on reasonable request (https://doi.org/10.17044/scilifelab.14229410 (accessed on 1 June 2022)).

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
