# Peer review of "Angiopoietin-2 Inhibition of Thrombomodulin-Mediated Anticoagulation—A Novel Mechanism That May Contribute to Hypercoagulation in Critically Ill COVID-19 Patients"

_biomedicines, 2022, doi:10.3390/biomedicines10061333_

Round 1

Reviewer 1 Report

Hultstrom and colleagues present an interesting draft entitled Angiopoietin-2 inhibition of thrombomodulin-mediated anticoagulation – a novel mechanism that may contribute to hypercoagulation in critically ill COVID-19 patients for consideration as an original article in the journal BioMedicines.

The introduction section is very clear, well-written and well-structured. It gives very good insight into the problem raised in the paper as well as all the cited references are solid and up-to-date. I have any concerns, even minor, regarding this part of the paper. Very similar feelings gave me the lecture of the Discussion section. I really appreciate how the Authors lead this section as well as how smoothly they discuss relevant literature for this area. 

Many majors come into play when analyzing results and methodlogy. Please let me point out my concerns:

Materials and Methods section is extensive and will allow reproduction of the described experiment, so it fits the state of the art. I am surprised that wt/wt mice have been used as a control group. When generating strains it is a good practice to use heterozygotes as controls. My main concern is also the span of the age of used mice - 8-20 is an extremely high span that might introduce lots of bias to the murine results. I would like to see how many mice have been used in total, with their sex- and age-based characteristics. This section must be cleared and the exact number of mice must be given.

An additional question comes to my mind regarding the bleeding time experiment. Firstly, I am surprised that the Authors did not include "pure" control mice since this technique has very high variability across the labs. Moreover, the volume, as well as time+volume calculations, are missing. There should be included these values since the bleeding time experiment has two main outcomes: how long (time) and how much (volume) of blood has been "collected". 

Additional serious red flag waves for Elisa results. It looks that the outcome is pretty heterogeneous, which does not generate any problem so far, but the used Elisa kits have a range of up to 3 and 4 ng/ml. The authors report much higher concentrations that are not possible to obtain when using described Elisa kits.

I would like to see a repetition of western data presented in Figure 1E - a group of ANGPT1 looks very heterogeneous - it should be repeated without doubt or this data should be completely removed since no believable conclusions can be withdrawn from it.  

Please describe in detail the methodology for lung tissue collection and harvesting since it is a missing part.

Please provide the methodology for ROTEM analysis. 

Figure 2A - most bar A values are outside the Elisa Kit range. 

Providing the confocal data showing colocalization of ANGPT2 and TM in lung epithelial (or other types of cells) would increase the power of the Author's conclusions. 

Please provide the patient's full morphology data as supplementary material.

Figure 5D - how it comes that there is no correlation between IL-6 and TNF-alpha? It looks questionable. 

Author Response

Hultstrom and colleagues present an interesting draft entitled Angiopoietin-2 inhibition of thrombomodulin-mediated anticoagulation – a novel mechanism that may contribute to hypercoagulation in critically ill COVID-19 patients for consideration as an original article in the journal BioMedicines.

The introduction section is very clear, well-written and well-structured. It gives very good insight into the problem raised in the paper as well as all the cited references are solid and up-to-date. I have any concerns, even minor, regarding this part of the paper. Very similar feelings gave me the lecture of the Discussion section. I really appreciate how the Authors lead this section as well as how smoothly they discuss relevant literature for this area.

We thank the reviewer for these positive notes.

Many majors come into play when analyzing results and methodology. Please let me point out my concerns:

  1. Materials and Methods section is extensive and will allow reproduction of the described experiment, so it fits the state of the art. I am surprised that wt/wt mice have been used as a control group. When generating strains it is a good practice to use heterozygotes as controls. My main concern is also the span of the age of used mice - 8-20 is an extremely high span that might introduce lots of bias to the murine results. I would like to see how many mice have been used in total, with their sex- and age-based characteristics. This section must be cleared and the exact number of mice must be given.

We realize that we should have been clearer in our explanations. In the current breeding our mice were bred with heterozygote parents: w/flox; Cdh5-creERT2 x w/flox; Cdh5-creERT2. Mice with flox/flox;Cdh5-creERT2 and wt/wt;Cdh5-creERT2 were induced (treated with tamoxifen) as described and used for study. Thus, the wt/wt;Cdh5-creERT2 controls are littermates to the flox/flox;Cdh5-creERT2 mutants (controlling for possible differences in genetic background), they carried Cre (controlling for possible unspecific Cre recombination activity/toxicity), and were treated with tamoxifen (controlling for possible “side effects” of tamoxifen. We do not understand the argument why it is “good practice” to use heterozygotes as controls as long as wildtypes are littermates and not from a parallel breeding program. From our experience heterozygous mice sometimes have a phenotype and we wanted to avoid this possible confounder. The breeding and experimental design of these experiments have now been better defined in Methods.

Concerning the age span, we do not think it is a problem as long as the phenotype is consistent and the age is the same for mutants and controls (as is the case here since we used littermates). The fact that the phenotype is consistent over a span of age in both sexes is rather an advantage, we think, and a testimony to the stability and relevance of the observed phenotype.  We have now clarified the age and sex distribution for the mice used in each experiment. Consistently controls (WT mice or IgG injected mice) have a bleeding time around 50-75 and we have not observed any sex or age influence on bleeding time.

  1. An additional question comes to my mind regarding the bleeding time experiment. Firstly, I am surprised that the Authors did not include "pure" control mice since this technique has very high variability across the labs. Moreover, the volume, as well as time+volume calculations, are missing. There should be included these values since the bleeding time experiment has two main outcomes: how long (time) and how much (volume) of blood has been "collected".

Here we are not sure what the reviewer means by ‘pure’ control mice. Is it that controls in i.p injections received human IgG? We think it is relevant to inject human IgG in controls when comparing to groups injected with human proteins (ANGPT2 and ANGPT1). This is commonly done to exclude that changes are caused by immune activation when injecting human proteins in mice. The reviewer has a point in that there is a large variability between labs when it comes to the tail-bleeding results, perhaps because every lab does things slightly different. The important thing, however, is that our controls and wildtype mice across several different experiments have similar results suggesting that we are performing tail-bleeding experiments in a highly reproducible way. Unfortunately, we did not collect the samples from tail bleeding so the blood loss in these experiments is not possible to estimate in retrospect. However, both the increase of plasma TAT and decrease of activated protein C support our findings that ANGPT2 injection leads to a more coagulative state, which is consistent with the tail bleeding result.

  1. Additional serious red flag waves for Elisa results. It looks that the outcome is pretty heterogeneous, which does not generate any problem so far, but the used Elisa kits have a range of up to 3 and 4 ng/ml. The authors report much higher concentrations that are not possible to obtain when using described Elisa kits.

As per laboratory practice, all samples were diluted to fit within the standard curve. As an example, human plasma samples were diluted 1:15 for ANGPT1 and ANGPT2 ELISAs and 1:300 for VWF and ADAMTS13 ELISAs.

  1. I would like to see a repetition of western data presented in Figure 1E – a group of ANGPT1 looks very heterogeneous – it should be repeated without doubt or this data should be completely removed since no believable conclusions can be withdrawn from it.

Here, we respectfully disagree. The slight heterogeneity (we object to the description of the results as “very heterogeneous”) is simply the nature of the obtained data and an honest representation of the experiment. Importantly, the difference is significant in spite of the slight heterogeneity as shown.

  1. Please describe in detail the methodology for lung tissue collection and harvesting since it is a missing part.

Please find that the methods part has been updated with a more detailed description for lung lysate preparation.

  1. Please provide the methodology for ROTEM analysis.

Please find that the methods part has been updated with a more detailed description of TEG.

  1. Figure 2A – most bar A values are outside the Elisa Kit range.

Please see reply for comment 3.

  1. Providing the confocal data showing colocalization of ANGPT2 and TM in lung epithelial (or other types of cells) would increase the power of the Author’s conclusions.

We agree with the reviewer that showing colocalization of ANGPT2 and TM by confocal imaging in addition to immunoprecipitation data would strengthen our conclusions. Thrombomodulin is expressed along the endothelial lumen as is Tie2. We have now included immunofluorescence staining showing ANGPT2 and thrombomodulin in lung endothelium (Figure 1e), however, we cannot exclude that ANGPT2 could be binding TIE2. We are exploring proximity ligation assay to show specific binding, but the technical challenges on tissue sections are well known and this is ongoing work.

  1. Please provide the patient’s full morphology data as supplementary material.

We do not have any morphology data from patients. For individual patient data we have the following statement in the manuscript: Individual level patient data can be made available on reasonable request (https://doi.org/10.17044/scilifelab.14229410).

  1. Figure 5D - how it comes that there is no correlation between IL-6 and TNF-alpha? It looks questionable.

While this result may not be according to the reviewer’s expectation, this is the data that we obtained. We have no specific explanation for the lack of correlation, but we also do not see this as a problem or that it would raise concerns about the validity of the data. The absence of a correlation between IL6 and TNF-alpha is also not inconsistent with the main data, message and conclusions form our study. 

Reviewer 2 Report

The authors investigated the mechanism of hypercoagulability and endothelial dysfunction in COVID-19 patients, and they showed that angiopoietin-2 has important role for the mechanism and it is also related to thrombomodulin by the analysis of both in vivo and in vitro. The content and approach are unique and interesting. However, there are some points required to revise. The detailed information was shown below.

1. The authors focused on the relationship between angiopoietin-2 and thrombomodulin and investigated that in vivo and in vitro. Although the title is clear, the explanation about thrombomodulin is too short and it is not easy to understand the relationship in the abstract for readers. Please revise the abstract and include the results of thrombomodulin.

2. The authors described the association with disseminated intravascular coagulation in conjunction with sepsis, and they also considered the relation between these mechanism and COVID-19. For the thrombosis mechanisms in COVID-19 patients, the following references are useful for readers’ understanding. Please cite the following references.

J. Thachil et al. ISTH DIC subcommittee communication on anticoagulation in COVID-19. J Thromb Haemost. 2020; 18: 2138-2144.

T. Iba et al. Coagulopathy in COVID-19. 2020; 18: 2103-2109.

3. A lot of thrombomodulin words are used in this manuscript. However, there are some types of thrombomodulin like full length thrombomodulin on endothelial cells, soluble thrombomodulin, recombinant thrombomodulin. Full length thrombomodulin has anticoagulant function on endothelial cells. Soluble thrombomodulin is the marker of endothelial cell injury, and it is considered that this form have low anticoagulant activity. Although recombinant thrombomodulin would have anticoagulant function, it is soluble form. Please define each form. Especially, please describe what kinds of thrombomodulin is used in the experiment.

4. Based on the difference among the types of thrombomodulin, please discuss the effect of increasing thrombomodulin concentration in plasma samples. In addition, there may be some limitations about the data interpretation if authors soluble recombinant thrombomodulin to use in vivo experiment. Please also describe this limitation in the discussion section.

5. The following sentence in line 214 looks like some thin wrong.

> Tail bleeding experiments in rodents are commonly used e to evaluate factors that affect the coagulation system

Please confirm.

6. The authors compared among a lot of markers in Figure 5. The data is interesting and there are some correlations between these markers. Especially, the unique markers like vWF and ADAMTS13 are included. This data is informative, please discuss the detail and the consideration in the discussion section.

Author Response

The authors investigated the mechanism of hypercoagulability and endothelial dysfunction in COVID-19 patients, and they showed that angiopoietin-2 has important role for the mechanism and it is also related to thrombomodulin by the analysis of both in vivo and in vitro. The content and approach are unique and interesting. However, there are some points required to revise. The detailed information was shown below.

We thank the reviewer for your time and feedback.

  1. The authors focused on the relationship between angiopoietin-2 and thrombomodulin and investigated that in vivo and in vitro. Although the title is clear, the explanation about thrombomodulin is too short and it is not easy to understand the relationship in the abstract for readers. Please revise the abstract and include the results of thrombomodulin.

Please find that the abstract has been updated to introduce thrombomodulin better.

  1. The authors described the association with disseminated intravascular coagulation in conjunction with sepsis, and they also considered the relation between these mechanism and COVID-19. For the thrombosis mechanisms in COVID-19 patients, the following references are useful for readers’ understanding. Please cite the following references. Thachil et al. ISTH DIC subcommittee communication on anticoagulation in COVID-19. J Thromb Haemost. 2020; 18: 2138-2144, Iba et al. Coagulopathy in COVID-19. 2020; 18: 2103-2109.

The suggested references have been added to the introduction.

3. A lot of thrombomodulin words are used in this manuscript. However, there are some types of thrombomodulin like full length thrombomodulin on endothelial cells, soluble thrombomodulin, recombinant thrombomodulin. Full length thrombomodulin has anticoagulant function on endothelial cells. Soluble thrombomodulin is the marker of endothelial cell injury, and it is considered that this form have low anticoagulant activity. Although recombinant thrombomodulin would have anticoagulant function, it is soluble form. Please define each form. Especially, please describe what kinds of thrombomodulin is used in the experiment.

The manuscript has been updated with better indications on when recombinant soluble thrombomodulin was used (i.e.  all in vitro experiments). The thrombomodulin from R&D system that we have used contain the domains necessary for thrombin binding and anticoagulation and a reference for this has been added. As the reviewer points out, it has been demonstrated that soluble thrombomodulin in plasma has lower anticoagulative properties (30-50%), compared to endothelial full length thrombomodulin. The reason for this is not known but it is possible that the activity is lower per se, or that an inhibitor is bound (like ANGPT2). This has been added to the discussion in the manuscript.

  1. Based on the difference among the types of thrombomodulin, please discuss the effect of increasing thrombomodulin concentration in plasma samples. In addition, there may be some limitations about the data interpretation if authors soluble recombinant thrombomodulin to use in vivo experiment. Please also describe this limitation in the discussion section.

As mentioned above, we have used a recombinant soluble thrombomodulin with thrombin and anticoagulative domains intact. In our in vitro experiment we utilized 20 nM ANGPT2 (and ANGPT1). This dose was guided by Daly et al (Sci Rep 2018) were 10 nM ANGPT1/ANGPT2 started to show inhibition of protein C activation in HUVEC cultures, and a robust response was seen when adding 100 nM ANGPT2. The doses used higher than described by us and other in pathological conditions. However, it is likely that ANGPT2 varies extensively among vessel types and location and is upregulated in response to angiogenic and inflammatory activation. It is therefore possible that inhibition of TM occurs only locally at sites of high ANGPT2 release. These limitations have been added to the discussion.

  1. The following sentence in line 214 looks like something wrong.

> Tail bleeding experiments in rodents are commonly used e to evaluate factors that affect the coagulation system

Please confirm.

Thanks, the sentence has been corrected in the manuscript.

  1. The authors compared among a lot of markers in Figure 5. The data is interesting and there are some correlations between these markers. Especially, the unique markers like vWF and ADAMTS13 are included. This data is informative, please discuss the detail and the consideration in the discussion section.

More detailed data for VWF and ADAMTS13 have now been added in Results, Table 3 and Suppl. Figure 2. A section about VWF and ADAMTS13 has been added to the Discussion.

Round 2

Reviewer 1 Report

The Authors adequately addressed all the raised issues as well as improved the draft substantially. I have no more comments.